# Surface Hydration Protects Cystic Fibrosis Airways from Infection by Restoring Junctional Networks

**DOI:** 10.3390/cells11091587

**Published:** 2022-05-09

**Authors:** Juliette L. Simonin, Alexandre Luscher, Davide Losa, Mehdi Badaoui, Christian van Delden, Thilo Köhler, Marc Chanson

**Affiliations:** 1Department of Cell Physiology & Metabolism, Faculty of Medicine, University of Geneva, 1211 Geneva, Switzerland; juliette.simonin@unige.ch (J.L.S.); losa.davide@gmail.com (D.L.); mehdi.badaoui@unige.ch (M.B.); 2Department of Microbiology and Molecular Medicine, Faculty of Medicine, University of Geneva, 1211 Geneva, Switzerland; alexandre.luscher@unige.ch (A.L.); christian.vandelden@hcuge.ch (C.v.D.); thilo.kohler@unige.ch (T.K.); 3Department of Medicine Specialties, Division of Infectious Diseases, Geneva University Hospitals, 1211 Geneva, Switzerland

**Keywords:** mucosal immunity, *P. aeruginosa*, airway surface liquid, epithelium integrity, cystic fibrosis

## Abstract

Defective hydration of airway surface mucosa is associated with recurrent lung infection in cystic fibrosis (CF), a disease caused by CF transmembrane conductance regulator (*CFTR*) gene mutations. Whether the composition and/or presence of an airway surface liquid (ASL) is sufficient to prevent infection remains unclear. The susceptibility to infection of polarized wild type and *CFTR* knockdown (CFTR-KD) airway epithelial cells was determined in the presence or absence of a healthy ASL or physiological saline. CFTR-KD epithelia exhibited strong ASL volume reduction, enhanced susceptibility to infection, and reduced junctional integrity. Interestingly, the presence of an apical physiological saline alleviated disruption of the airway epithelial barrier by stimulating essential junctional protein expression. Thus, rehydrated CFTR-KD cells were protected from infection despite normally intense bacterial growth. This study indicates that an epithelial integrity gatekeeper is modulated by the presence of an apical liquid volume, irrespective of the liquid’s composition and of expression of a functional CFTR.

## 1. Introduction

Mucociliary clearance provides an effective defense mechanism for the removal of inhaled particles and pathogens from airways. This defense mechanism is altered in people with cystic fibrosis (CF), a lethal automosal recessive genetic disease caused by mutations of the CF transmembrane conductance regulator (*CFTR*) gene encoding the CFTR chloride channel. People with CF develop severe respiratory disorders characterized by chronic bacterial infections concomitant with exacerbated inflammation and luminal tracheo-bronchial obstruction, leading to progressive lung function deterioration [1,2].

Optimal mucociliary clearance depends on the volume and composition of the airway surface liquid (ASL), and hence on the continuous regulation of airway surface hydration [3,4]. The ASL is comprised of a periciliary fluid and an overlaying layer of mucins [5]. In CF, loss of chloride secretion across the apical membrane of airway epithelial cells results in ASL dehydration, thereby increasing the mucin concentration within the periciliary layer, reducing cilia beating and thus impairing mucociliary clearance [6,7,8]. It is well reported that ASL composition in terms of ion concentration, pH, and antimicrobial peptides contributes to innate immunity through bacterial killing [9,10,11,12,13,14,15]. These key defense functions are, however, altered in CF. An intact epithelial barrier is also essential to avoid bacteria transmigration in the underlying tissues and their access to nutritious and replicative niches as well as to limit the diffusion of bacterial toxins [16,17,18]. Opportunistic bacteria such as *Staphylococcus aureus* and *Pseudomonas aeruginosa*, the main morbidity-causing pathogens in CF, have been shown to target tight- and adherens-junctions [19,20,21,22,23], demonstrating that ASL homeostasis contributes to bacterial clearance and airway epithelial defense.

The relationship between ASL and airway epithelium junctional integrity has been addressed in a few studies [24,25,26,27]. Investigations on pulmonary epithelial cell lines have revealed that air–liquid interface conditions adjust claudin-1 expression via retinoic acid signaling [28]. Another study showed that washing away ASL components is prevented by erbB1-mediated formation of tight-junctions in primary cultures of human airway epithelial cells [29]. CFTR may also regulate paracellular permeability of the airway epithelium via protein–protein interaction involving the scaffold protein ZO-1 at the tight-junctions [30] or secretion of the tight-junction protective molecule lipoxin A4 [31]. It remains unclear, however, whether the biophysical presence alone of an apical liquid is able to prevent bacteria-induced cytotoxicity during the infection of CFTR-deficient airway epithelial cells.

We show here that *P. aeruginosa*-induced cytotoxicity of CFTR-deficient airway epithelial cells is reduced by the presence of a liquid on an apical surface, irrespective of its composition or expression of a functional CFTR at the apical membrane. These results provide a new paradigm for CF etiology, whereby an epithelial integrity gatekeeper is modulated by changes in apical surface hydration.

## 2. Materials and Methods

### 2.1. Cell Culture

Human airway epithelial Calu-3 cells were purchased from the American Type Culture Collection (ATCC^®^ HTB-55™, Manassas, VA, USA). CFTR-CTL and CFTR-KD cells expressing the wild type CFTR and knocked down for the *CFTR* gene, respectively, were generated from Calu-3 cells by CRISPR-Cas9 [32]. Cells were cultured in Minimum Essential Medium (MEM) GlutaMAX™ (Gibco 41090-28, Waltham, MA, USA) supplemented with 10% heat-inactivated Fetal Bovine Serum (FBS, Gibco 10270-106, Waltham, MA, USA), 1% non-essential amino acids 100X (Gibco 11140-035, Waltham, MA, USA), 1% HEPES 1M (Gibco 15630-056, Waltham, MA, USA), 1% sodium pyruvate 100X (Gibco 11360-039, Waltham, MA, USA), 1% penicillin/streptomycin, and 0.25 µg/mL fungizone (BioConcept 4-02F00-H, Postfach, Switzerland) at 37 °C in humidified 5% CO_2_. The medium was changed every two days.

Well-polarized monolayers of CFTR-CTL and CFTR-KD Calu-3 cells were obtained by seeding 1.75 × 10^5^ cells onto 0.33 cm^2^ porous (0.4 µm) Transwell polyester inserts (Transwell 3470, Corning Life Sciences, Hazebrouck, France) and cultured for 5 days under submerged conditions when cells reached confluence (around 3.3 × 10^5^ cells in one 0.33 cm^2^ filter, ≈10^6^ cells/cm^2^). Epithelial polarization was optimized by culture at the air–liquid interface (ALI) for 15–20 days. For the infection experiments, penicillin, streptomycin, and fungizone were removed at least one week before infection to allow for the secretion of an ASL volume without any antibiotic and antifungal medication. After ASL removal, new apical secretion was observed in the CFTR-CTL cultures from the first day onward and reached the maximal volume within one week.

### 2.2. Apical Surface Liquid Manipulation

CFTR-CTL epithelia were exposed to different apical conditions by maintaining their natural ASL (CTL-ASL) vs. removing its CTL-ASL vs. removing and replacing its CTL-ASL with the same volume of a physiological saline (NaCl 154 mM, HEPES 10 mM, CaCl_2_ 1.2 mM): CTL-ASL vs.−CTL-ASL vs. + saline for 24 h, 48 h, and 96 h. HEPES was added to the physiological saline to provide pH-buffering capacity during the TEER measurement. In parallel, we exposed CFTR-KD epithelia to different apical conditions by maintaining their natural ASL (KD-ASL) vs. adding CTL-ASL from one CFTR-CTL culture vs. adding the same volume of the physiological saline: KD-ASL vs. +CTL-ASL vs. +saline for 24 h, 48 h, and 96 h. No apical washing was performed after removing the CTL-ASL or before the CTL-ASL vs. saline addition. ASL manipulation was performed on well-polarized CFTR-CTL and CFTR-KD cells at the ALI for 15–20 days.

For the infection experiments, ASL manipulations were performed 24 h (Δ*lasR*) or 48 h (Δ*fliC* and PAO1 WT) before inoculation with the bacterial strains. Liquids were kept during infections (except for “removed” conditions from the experiments in Appendix A) for a total liquid incubation of 48 h–72 h.

### 2.3. Bacteria and Infection

All experiments were performed with the *Pa* laboratory strain PAO1 [33] or with the PAO1-derived mutant strains Δ*lasR* (PAO1 *lasR*::Tc, deficient in *las* QS system [33]), Δ*fliC* (PAO1 *fliC*::Gm, non-motile [34]), and Δ*wbpL* (PAO1 *wbpL*::Gm, A-band, and B-band negative mutant [35]). Bacteria were grown in lysogeny broth (LB) at 37 °C with agitation (240 rpm).

Polarized CFTR-CTL and CFTR-KD Calu-3 cultures were infected as previously described [36]. Briefly, bacteria were grown overnight, washed, and resuspended in physiological saline at a density of 10^5^ CFU/mL. Ten µL of this suspension was then added apically to the CFTR-CTL and CFTR-KD cultures on Transwell filters (final inoculum of 10^3^ CFU per well, MOI of 0.003). Next, 10 µL of saline was added to the uninfected control cells. Cells were incubated for 6 h, 16 h, and 24 h at 37 °C.

To count apical CFU, 200 µL of physiological saline was added on top of the CFTR-CTL and CFTR-KD cultures and 100 µL was then collected for counting. Basolateral CFU counts were performed by collecting 200 µL of basal medium. Five µL of 10-fold serial dilutions in physiological saline were spotted on LB–agar plates and CFU counted after 24 h of incubation at 37 °C.

### 2.4. Epithelium Integrity

Epithelium integrity was evaluated in the CFTR-CTL and CFTR-KD epithelia grown at ALI for at least 15 days through the measurement of two parameters.

The area of lesion induced by the bacteria was estimated using microscopy. Since the CFTR-CTL and CFTR-KD cells express GFP [32], the area of lesion induced by the bacteria was estimated with fluorescence microscopy. For each Transwell insert, nine images were acquired by ImageXpress XL (Molecular Devices, San Jose, CA, USA) with a 4× objective. Pictures of the whole insert were reconstructed using MetaXpress^®^ software and the lesion areas were measured using ImageJ with the plugin “Stitching/Grid Collection stitching”. Data were expressed as % of the intact surface ((filter surface − lesion area)/filter surface).

The transepithelial resistance (TEER) values were measured using a voltmeter (EVOM, World Precision Instruments, Inc., Friedberg, Germany). TEER values were rapidly measured after removing ASL or saline from the filters and adding 200 µL of PBS or physiological saline at the apical side and 600 µL at the basal side of each insert. TEER measurements were performed in duplicate for each filter.

### 2.5. Western Blot

Proteins were extracted from the polarized CFTR-CTL and CFTR-KD Calu-3 cells using the Nonidet-P40 lysis buffer (150 mM NaCl, 50 mM Tris (pH 8.0), 1% NP-40 (AppliChem A1694, Milano, Italy)) and Roche complete™ *Protease Inhibitor* Cocktail (Cat. No. 04693124001, Sigma, St. Louis, MO, USA). The cell lysates were centrifuged at 10,000 *g* for 15 min at 4 °C and the protein concentration was quantified with a Pierce BCA Protein Assay Kit (ThermoFisher, Cat. No. 23228, Rockford, IL, USA). Amounts of 5–10 µg of the proteins were separated in denaturing SDS-PAGE gels (Bio-Rad 161-0301, Marnes-la-coquette, France), transferred onto a Porablot NCP nitrocellulose membrane (Macherey-Nagel, Cat. No. 741280, Hoerdt, France), and blocked for 1 h at room temperature (RT) using 3% BSA (Sigma-Aldrich A7906, St. Louis, MO, USA) in PBS–Tween (PanReac Applichem A4974, Barcelona, Spain) buffer. Primary antibodies (listed in Appendix A) were incubated with the membranes overnight at 4 °C with agitation. GAPDH (Merck, Darmstadt, Germany) or β-actin (Sigma, St. Louis, MO, USA) antibodies were used as the loading control. After primary antibody fixation (E-cadherin and β-catenin: Cell Signaling, Danvers, MA, USA; Claudin-3, claudin-2 and α-catenin: Abcam, Waltham, MA, USA; Occludin and ZO-1: Invitrogen, Rockford, IL, USA), the membranes were washed with PBS–Tween buffer, followed by horseradish peroxidase-coupled secondary antibody incubation. Proteins were finally detected using the chemiluminescent HRP substrate Immobilon™ Western (Millipore, Cat. No. WBKLS0500, Darmstadt, Germany) and quantified with ImageJ.

### 2.6. Confocal Microscopy

Polarized CFTR-CTL and CFTR-KD Calu-3 cells were fixed using 4% PFA (Sigma 158127, St. Louis, MO, USA) for 15 min at RT and permeabilized for 15 min at RT with 0.2% Triton 100 X (Sigma T-8787, St. Louis, MO, USA) buffer. The non-specific sites were blocked with PBS-BSA 3% solution for 30 min at RT and samples were then incubated with primary antibody targeting ZO-1 (Appendix A) at 4 °C overnight. After 3 × 5 min PBS washing, a goat anti-rabbit secondary AlexaFluor^®^ 647 antibody (ThermoFisher, Cat. No. A-21245, Rockford, IL, USA) was applied for 1 h at RT for the detection of the ZO-1 protein, while DAPI (Applichem, Cat. No. A4099, Milano, Italy) was used for the nuclear counterstaining. The fluorescence images and Z-stack were acquired with an LSM700 confocal microscope and ZEN software (ZEISS). The images were analyzed using ZEN and ImageJ software.

### 2.7. Short-Circuit Current

Transwell inserts were mounted on Ussing chambers (Physiologic Instruments, Reno, NV, USA). The short-circuit current (Isc) was recorded using a VCC MC6 amplifier (Physiological Instruments). Data were acquired using the interface DI-720 (DataQ Instruments) and Acquire & Analyze software 2.3 (Physiological Instruments, Reno, NV, USA).

Basal chambers were filled with a Krebs solution (in mM: 115.5 NaCl, 25 NaHCO_3_, 2.4 K_2_HPO_4_, 0.4 KH_2_PO_4_, 1.2 CaCl_2_-2H_2_O, 1.2 MgCl_2_-6 H_2_O, and 10 glucose) and apical chambers with a low Cl^-^ Krebs solution (in mM: 100.5 NaGluconate, 15 NaCl, 25 NaHCO_3_, 2.4 K_2_HPO_4_, 0.4 KH_2_PO_4_, 1.2 CaCl_2_-2 H_2_O, 1.2 MgCl_2_-6 H_2_O, and 10 glucose). Basal and apical chambers were gassed with 95% O_2_, 5% CO_2_ at 37 °C. The transepithelial potential difference was voltage-clamped at zero and the resulting Isc was recorded using a V/I clamp. The epithelial sodium channel (ENaC)-mediated currents were blocked by the addition of 100 µM amiloride (Millipore 129876, Darmstadt, Germany) at the apical side. cAMP-induced currents including CFTR-mediated currents were stimulated by the addition of a cocktail of forskolin (10 µM, Sigma F6886, St. Louis, MO, USA) and IBMx (100 µM, Sigma I5879, St. Louis, MO, USA) added to both chambers. GlyH-101 (20 µM, Merck 219671, Darmstadt, Germany) was added to the apical side to specifically inhibit CFTR activity. Then, bumetanide (100 µM, Sigma B-3023, St. Louis, MO, USA) was added to the basal side to inhibit Na^+^/K^+^/2 Cl^−^ co-transport activity. Finally, MgATP (100 µM, Sigma A9187, St. Louis, MO, USA) was added to the basal side. Transepithelial resistance was measured at the beginning and at the end of each experiment.

### 2.8. RNA Extraction, RT-PCR, and qPCR

Total RNA was extracted from the polarized Calu-3 cells with an RNeasy Mini Kit (Qiagen, Cat. No.74106, Hilden, Germany). RNA concentration and purity were verified with a Nanodrop 2000 (ThermoFisher, Rockford, IL, USA) spectrophotometer. Genomic DNA was removed using the gDNA wipeout buffer for 2 min at 42 °C and cDNA was synthetized with the QuantiTect Reverse Transcription Kit (Qiagen, Cat. No. 205311, Hilden, Germany). qPCR was performed with PowerUp™ SYBR™ Green Master Mix (Appliedbiosystems, Cat. No. A2574, Bedford, MA, USA) using the *StepOnePlus* Real-Time PCR system. Primer pairs (Microsynth, Balgach, Switzerland) for Claudin-3, occluding, ZO-1, E-cadherin, and β-catenin are shown in Appendix A. mRNA expression is represented as the absolute value (2^−ΔCt^) normalized to 18 S expression.

### 2.9. NanoString Gene Expression

The expression of the 579 human immune genes was measured in uninfected and Δ*lasR*-infected epithelial cells using Nanostring technology (NanoString^®^). Briefly, 100 ng of RNA from the Calu-3 cell lines were hybridized for 20 h at 65 °C with immune pathway probes (nCounter inflammation panel Human v2, NanoString^®^). Post-hybridization washing and bound-RNA loading on the nCounter Prep station were processed following NanoString^®^ guidelines. Sample normalization was performed on 13 housekeeping genes: *GUSB*, *HPRT1*, *TUBB*, *GAPDH*, *ABCF1*, *EEF1 G*, *G6 PD*, *OAZ1*, *POLR2 A*, *PPIA*, *RPL19*, *SDHA*, and *TBP*. A two-fold change was considered significant.

### 2.10. Statistical Analysis

Values were represented as mean ± SEM. The statistical tests were conducted using SigmaStat (Systat Software, Inc., San Jose, CA, USA) or GraphPad Prism software. The differences between the two groups were analyzed by the paired Student’s *t*-test while the differences between more than two groups were tested using the two-way analysis of variance (ANOVA) followed by Holm–Sidak post hoc tests. *p* < 0.05, *p* < 0.01, and *p* < 0.001 were considered significant and represented as *, **, and ***, respectively. NS indicates non-significant differences. *N* defines the number of performed experiments and *n* defines the number of technical replicates for each experiment.

## 3. Results

### 3.1. CFTR-KD Calu-3 Cells Exhibit Enhanced Susceptibility to Pa Virulence

The *CFTR* knockdown by CRISPR-Cas9 in the Calu-3 submucosal gland airway epithelial cell line (CFTR-KD) has previously been described [32]. Both CFTR-KD cells and their control counterpart (CFTR-CTL cells) were grown on Transwell inserts at an air–liquid interface (ALI). To evaluate their response to infection, we monitored the epithelial integrity of the CFTR-CTL and CFTR-KD cell cultures after the apical inoculation of 10^3^ CFU of the wild type PAO1 for 6, 16, and 24 h. The PAO1 induced wounds of the airway epithelium formed by the CFTR-CTL or CFTR-KD cells within 16 h, as evaluated by measurement of the injured area. After 24 h in the presence of PAO1, the CFTR-KD epithelium was entirely damaged (Figure 1A). As expected, the transepithelial resistance value (TEER), an indicator of junctional integrity, decreased more rapidly in the CFTR-KD cells compared to the CFTR-CTL cells (Figure 1B). The PAO1 growth was also determined in the apical (Figure 1C) and basolateral (Figure 1D) compartments. PAO1 proliferated in the apical compartments, resulting in a 5-log and 7-log increase in CFU count in the CFTR-CTL and CFTR-KD cultures, respectively, 24 h post-infection. The measurement of the bacterial amount in the basolateral compartments informed on the PAO1 ability to cross the airway epithelium. As shown in Figure 1D, PAO1 was detected in the basolateral compartment only 24 h post-infection. Thus, despite an intense bacterial proliferation, both the CFTR-CTL and CFTR-KD cultures were able to maintain some epithelial integrity until they reached irreversible damage at 24 h. The higher bacterial counts and the faster degradation of epithelial integrity suggest that cells lacking functional CFTR were more vulnerable to infection than the control Calu-3 cells.

To confirm the vulnerability of the CFTR-KD cells regarding CF context, we performed additional infections with three PAO1 mutant strains, deleted for specific virulence factors commonly found in CF patients [37,38]. We compared the effects of these mutant strains of attenuated virulence between the CFTR-CTL and CFTR-KD cells. Therefore, we monitored the injured area, TEER, apical, and basolateral growth of the wild type PAO1, quorum sensing (QS)-deficient mutant Δ*lasR*, flagella-deficient mutant Δ*fliC*, and outercore LPS deficient mutant Δ*wbpL* at 6, 16, and 24 h post-infection. As shown in Figure 2A, the integrity of the CFTR-CTL cells was not affected by the PAO1 mutants after 24 h of infection. In contrast, the CFTR-KD epithelium was destroyed in the presence of Δ*lasR* and Δ*fliC* and severely damaged by the Δ*wbpL* mutant. These results were confirmed by quantification of the injured area (Figure 2B), TEER (Figure 2C), apical (Figure 2D), and basolateral (Figure 2E) bacterial counts. Again, apical bacterial proliferation was higher in the CFTR-KD cell environment, with the detection of Δ*lasR* and Δ*wbpL* strains in the basolateral compartment. As expected for a mutant without flagella, Δ*fliC* bacteria were not detected in the basal compartment. Quantifications at 6 and 16 h are shown in Appendix A.

These results indicate the enhanced vulnerability of Calu-3 cells lacking a functional CFTR to PAO1, even to PAO1 mutants silenced for *fliC* and quorum-sensing signal receptor *lasR* genes.

### 3.2. CFTR-KD Calu-3 Cells Exhibit Altered Expression of Junctional Proteins

To identify a potential cause for the enhanced susceptibility of Calu-3 cells lacking a functional CFTR to infection, we studied the expression of junctional protein complexes in the CFTR-CTL and CFTR-KD cells by Western blot. Interestingly, we observed a decreased expression in proteins that are components of tight-junctions and adherens-junctions. Representative Western blots for claudin-3, claudin-2, and occludin as well as E-cadherin, β-catenin, and α-catenin are shown in Figure 3A,B for the tight-junction and adherens-junction proteins, respectively. Quantification revealed aa strong decreased expression for all proteins of these junctional complexes except for occludin (Figure 3C). However, no difference in claudin-3, occludin, ZO-1, E-cadherin, and β-catenin mRNA expression was observed between the CFTR-CTL and CFTR-KD cultures (Appendix A). Figure 3D shows a Western blot to confirm the decreased expression of CFTR in the CFTR-KD cells. In contrast to the polarized Calu-3 cells, the CFTR-CTL and CFTR-KD cells grown as monolayers did not show differences in the tight- and adherens-junction proteins (Appendix A).

Thus, the results suggest that loss of a functional CFTR causes defects in the polarization process, leading to alteration in the barrier integrity of the airway epithelium via degradation of some junctional proteins.

### 3.3. Rehydration of the CFTR-KD Calu-3 Cell Surface Restored Barrier Integrity

The CFTR chloride currents drive water movement to hydrate the airway surface liquid (ASL), with ASL dehydration being a feature of CF disease. The CFTR-KD Calu-3 cells recapitulate this feature as determined by the short-circuit current (Isc) measurements in Ussing chambers (Figure 4A,B). Figure 4A shows representative Isc traces of the CFTR-CTL and CFTR-KD cells consecutively treated with amiloride, a forskolin/IBMX cocktail, GlyH-101, bumetanide, and MgATP; quantification of Isc responses is shown in Figure 4B. As previously described, amiloride-sensitive ENaC-channels were absent in the Calu-3 cells [39]. The CFTR-CTL cells exhibited a solid response to the cAMP-cocktail that was blocked by the CFTR activity inhibitor GlyH-101 and further decreased by the inhibition of the NKCC cotransporter with bumetanide. In contrast, no cAMP-induced nor GlyH-101 responses were observed in the CFTR-KD cells; however, the cells responded to calcium purinergic stimulation with MgATP, which is likely to be mediated by intracellular calcium elevation. Calu-3 cells are known to produce efficiently apical secretion. Typically, an ASL volume (hereafter CTL-ASL) of about 30 µL is spontaneously generated by CFTR-CTL cells cultured at ALI for 15 days, reaching up to 60–100 µL after 3 weeks; in contrast, the CFTR-KD cells produced almost no apical liquid, whose volume was not quantifiable without microscopic tools (Figure 4C). In addition, the *CFTR*-silenced Calu-3 cells producing a low ASL volume (KD-ASL) showed a marked decrease in their TEER compared to the Calu-3 cells with a CTL-ASL after 3 weeks at ALI (Figure 4D).

To investigate a potential link between ASL and junctional integrity, we compared the TEER of the CFTR-CTL and CFTR-KD cultures after manipulation of the ASL, as illustrated in Figure 5A. The CTL-ASL from 3 week-old CFTR-CTL cell cultures was removed (-ASL) and transferred onto the surface of the CFTR-KD cultures for 24, 48, and 96 h (Figure 5B–D). The effects of removing/adding CTL-ASL was also compared with that of a similar volume of physiological saline. The removal of CTL-ASL decreased the TEER in the CFTR-CTL cells at all-time points studied, an effect that was prevented by the apical addition of saline. Interestingly, the TEER of the CFTR-KD cultures progressively increased with the time of incubation with either CTL-ASL or saline from 48 h, and reached similar values with CTL-ASL than the control CFTR-CTL cultures at 96 h of incubation.

To determine whether the improved TEER in the CFTR-KD cultures treated with CTL-ASL or saline was associated with changes in the integrity of epithelium barrier, we studied the expression of tight-(claudin-3, claudin-2, occludin) and adherens-(E-cadherin, β-catenin, α-catenin) junction proteins by Western blot. A series of experiments at 24 h and 48/96 h of CTL-ASL or saline incubation was performed on the CFTR-KD cultures; representative Western blots are shown in Figure 6. Compared to the KD-ASL condition, the transfer of CTL-ASL or saline to the CFTR-KD cells increased the expression of α-catenin, β-catenin, E-cadherin, and claudin-3 (Figure 6), with α-catenin first being markedly reestablished by rehydration. Moreover, it has been reported that CFTR interacts with ZO-1 [24], a tight junction scaffolding protein also associated with adheren-junctions [40]. Thus, we studied the expression and localization in polarized CFTR-CTL and CFTR-KD epithelia with confocal imaging ZO-1. As shown in Appendix A, the typical apical junctional network formed by ZO-1 in CFTR-CTL was severely disrupted in the CFTR-KD cells. Likewise, transfer of CTL-ASL or saline to the CFTR-KD cultures increased within 24 h of ZO-1 expression, which showed a membrane organization similar to that observed in the CFTR-CTL cells.

Thus, a liquid presence on the apical surface is critical to maintaining the integrity and barrier function of the airway epithelium.

### 3.4. Rehydration of the CFTR-KD Calu-3 Cell Surface Reduced the Epithelium Vulnerability to Infection

The Δ*fliC* and Δ*lasR* PAO1 mutant strains with attenuated virulence destroyed the CFTR-KD epithelium within 24 h (Figure 2). To evaluate the level of protection offered by CTL-ASL or saline to infection, we studied the effects of Δ*fliC* and Δ*lasR* inoculation on the injured area and TEER of the CFTR-CTL and CFTR-KD epithelia, exposed or not to apical hydration. As Δ*lasR* expressed the strongest virulence in the CFTR-KD cells compared to Δ*fliC* (Figure 2 and Appendix A), we first evaluated the effect of apical hydration against Δ*fliC*. For these experiments, CTL-ASL or saline was added for 24 h–48 h before inoculation with 10^3^ CFU Δ*fliC* for 24 h. Importantly, CTL-ASL or saline protected the CFTR-KD epithelium from injury by Δ*fliC* (Figure 7A). This protection was associated with increased TEER (Figure 7D). Of note, removing CTL-ASL from CFTR-CTL cells decreased the TEER of infected cells, although with no significant impact on the damaged epithelium surface; this decrease in TEER was less pronounced in the CFTR-CTL cells that received saline apically. We next evaluated whether apical hydration was still able to protect epithelium from infection with the more virulent Δ*lasR* strain. To do so, we performed infections with 10^3^ CFU Δ*lasR* for 16 h and 24 h. At 16 h post-infection, Δ*lasR* already induced damage on the CFTR-KD epithelium and a strong TEER decrease (Appendix A and Figure 7B,E). Despite this virulence, epithelial damage was prevented by CTL-ASL incubation and reduced by saline incubation (Figure 7B). Both the CTL-ASL and saline conditions were associated with increased TEER (Figure 7E). At 24 h post-infection, the macroscopic epithelial integrity was still preserved in both the hydrated CFTR-KD cell conditions (Figure 7C and Appendix A) while more variability was observed in the TEER measurements (Figure 7F). The manipulation of the ASL in both the CFTR-CTL and CFTR-KD cells did not affect the bacterial proliferation capacities as determined by the apical CFU counts of Δ*fliC* (Figure 7G) and Δ*lasR* (Figure 7H,I) strains. Finally, rehydration reduced the bacterial amount detected in the basolateral compartments of CFTR-KD cells infected with Δ*lasR* (Appendix A).

Finally, we evaluated whether apical hydration could offer protection against the WT PAO1 strain 16 h post-infection. We observed that both CTL-ASL and saline were able to protect the CFTR-KD epithelium from bacterial-induced lesions (Figure 8A,D). This protection was associated with an increased TEER (Figure 8B), while apical bacterial growth remained (Figure 8C).

ASL manipulation introduced a dilution factor that could explain the reduced virulence of the PAO1 mutant strains. To address this possibility, CFTR-CTL and CFTR-KD cells were treated as above but CTL-ASL was removed just before infection with Δ*fliC* for 24 h. Despite the absence of an ASL during infection, this handling reproduced the beneficial effects of CTL-ASL pretreatment in terms of the epithelium lesion and TEER changes, although values were slightly lower (Appendix A).

ASL manipulation may also affect other actors of the innate defense mechanisms of airway epithelial cells by increasing bacterial killing or secreting inflammatory cytokines and chemokines. In another series of control experiments, we first tested the effects of conditioned saline on PAO1, Δ*fliC*, and Δ*lasR* proliferation. To this end, the apical surface of the CFTR-CTL and CFTR-KD epithelia was washed and saline was added for 4 h to collect secretions. The conditioned salines (hereafter CTL-saline and KD-saline) were recovered and infected with 10^5^ CFU bacteria for 24 h. As shown in Appendix A, PAO1, Δ*fliC,* and Δ*lasR* were not able to grow in a saline that had not been in contact with the epithelial cells. In contrast, all bacteria strains could grow in the conditioned salines and even showed higher proliferation in the KD-saline. To gain insights into the immune response of the CFTR-CTL and CFTR-KD cells to ASL manipulation, we used a multiplex gene expression analysis with a panel of 579 immunology related genes. Relative changes of gene expression in response to ASL manipulation followed or not by infection with the 10^5^ CFU of Δ*lasR* strain for 16 h were normalized to uninfected CFTR-CTL in the CTL-ASL condition (Appendix A) or CFTR-KD in the KD-ASL condition (Appendix A). CTL-ASL removal from the CFTR-CTL epithelia or addition of CTL-ASL or saline to CFTR-KD epithelia did not change the basal level of immune gene expression. In contrast, infection with Δ*lasR* induced an immune response, but exhibited a similar profile regardless of the presence or absence of a liquid at the apical side. Appendix A shows all gene expression changes in response to ASL manipulation in the uninfected and infected CFTR-CTL and CFTR-KD cells.

Thus, these results indicate that, despite the remaining intense bacterial proliferation, apical surface hydration allowed for efficient protection against bacterial infections by maintaining the junctional integrity of the epithelium, and not only by the dilution of bacteria number and/or by changes in the epithelial immune profile.

## 4. Discussion

In the present study, we investigated whether the presence of a liquid volume on apical surface protected the CF airway epithelium from *Pa*-induced cytotoxicity. Pathogens including *Pa* [16] use a number of strategies to invade and traverse epithelial barriers at cell–cell junctions. Wild type PAO1 destroyed within 24 h the epithelia formed by ALI cultures of the Calu-3 airway epithelial cells. Mutations in *lasR*, *fliC*, and *wbpL* genes are commonly found in adapted *Pa* isolates from CF airways [38,41]. By comparing the wild type PAO1 to the Δ*lasR*, Δ*fliC,* and Δ*wbpL* mutant PAO1 strains, we pointed to *Pa* virulence factors that triggered the loss of tissue integrity in this cell model. Indeed, Δ*lasR*, Δ*fliC*, and partially Δ*wbpL* PAO1 mutants were harmless in the CFTR-CTL epithelium, indicating that the accumulation of extracellular quorum-sensing signals and activation of epithelial Toll-like receptors (TLRs) are primarily responsible for short-term (<24 h) epithelial damage. More specifically, first, the quorum sensing homoserine lactone C12 was shown to alter the airway epithelium integrity [22,42,43]. Second, flagellin and LPS, activating TLR5 and TLR4, respectively, trigger intracellular signaling cascades to elicit innate immune responses, amenable to modulating the junctional permeability during the transmigration of immune cells [44,45]. Here, we showed that the deletion of these virulence factors was sufficient to prevent the integrity loss of CFTR-CTL epithelia, but not of CFTR-KD. Consistent with reported observations [12,13,31], wild type and mutant PAO1 strains in the CFTR-KD epithelia showed a higher ability to traverse the epithelium as well as higher apical proliferation, suggesting that KD-ASL is unable to kill bacteria. These observations indicate that the CFTR-KD Calu-3 cell model recapitulates vulnerability to infection of the dehydrated CF airway epithelium.

When grown at ALI, the CFTR-KD cells failed to produce a measurable ASL volume in contrast to the CFTR-CTL cells. In the ASL transfer experiments, the presence of this healthy apical environment preserved the integrity of the CFTR-KD epithelia infected with PAO1 mutant strains. Interestingly, similar protection of the CFTR-KD epithelia against PAO1 virulence was achieved when a physiological saline was used instead of CTL-ASL. It has been proposed that CF ASL biochemical anomalies (low bicarbonate, acid ASL, high salt, …) impaired antibacterial peptide activity [12,13,46,47] and/or compromised mucin expansion, thereby reducing bacteria trapping [6]. In our study, higher proliferation of PAO1, Δ*lasR*, and Δ*fliC* was still observed in the physiological saline conditioned with secretions collected from the CFTR-KD epithelia compared to the CFTR-CTL cultures. As physiological saline is a bicarbonate-free, HEPES-buffered-solution, these observations indicate that factors other than ASL composition and bacterial killing capacity contributed to the integrity of the CFTR-KD epithelium infected with PAO1 mutant strains.

A link between CFTR expression and airway epithelium junctional permeability has been previously established [24,25,26,27]. *CFTR* silencing reduced ZO-1 expression, while CFTR co-localized with the scaffold protein at the tight-junctions, suggesting that CFTR may regulate their assembly through protein–protein interactions [24,27]. It is also well established that the PDZ-domain containing adaptor molecules such as NHERF1/EBP50 and NHERF2 associate with the CFTR C-terminal PDZ-binding motif and tether CFTR to the actin cytoskeleton [24,48]. Indeed, NHERF1 overexpression in CFBE41o-cells rescued CFTR-dependent chloride secretion by stabilizing F508del CFTR on the apical membrane [49]. Thus, dysfunctional CFTR may alter the cytoskeleton and organization of junctional complexes during polarization. Consistently, altered and/or disorganized expression of tight- and adherens-junction proteins have been reported in various CF epithelium models [24,25,26,27]. We also observed that claudin-3, E-cadherin, α-, and β-catenin decreased expressions as well as discontinuous ZO-1 membrane localization in the CFTR-KD epithelium. Occludin is one of the tight junction proteins that was not affected, at least at the time points studied. Interestingly, the decreased protein expression appeared to be due to protein degradation rather than decreased transcription. This observation suggests that junctional proteins may be degraded by different mechanisms or that degradation occurred at different rates. Importantly, the transfer, not only of a healthy ASL but also of physiological saline, recovered the expression of these proteins in the CFTR-KD cells, excluding a role of CFTR-dependent protein interaction in this process. In addition, restoring junctional complexes in the CFTR-KD cells by the addition of an apical liquid volume is sufficient to preserve the epithelium integrity during infection with PAO1 mutant strains, a protection that was maintained even when ASL was removed prior to infection. Finally, the manipulation of ASL did not modify the expression of inflammatory genes under basal or infected conditions. Although in vivo confirmation is awaited, these observations indicate that an epithelial integrity gatekeeper is modulated by the presence of an apical liquid volume. The latter modulation may occur independently of the presence of a functional CFTR at the apical membrane.

CF airway rehydration was recently shown to recover mucus properties without restoring pH value or bicarbonate concentration in bronchial epithelial cells from subjects with G551D or F508del mutations exposed to CFTR modulators [50]. Consistent with our findings, apical surface rehydration of the CF airway epithelium may reinforce epithelial barrier integrity. It is possible that junctional complexes may sense changes in the CF apical environment, triggering mechanical signals that would regulate cell–cell contact [51]. Alternatively, CFTR may participate in the mechanosensing of changes in the ASL volume, hydrostatic pressure, and/or membrane stretching [52,53,54]. Although further studies are needed to unravel the precise mechanisms by which a liquid presence on an epithelium regulates its barrier integrity, our results emphasize alternative therapies to rehydrate the CF airway surface for *CFTR* mutations, not rescued by CFTR modulators. They also support rehydration therapy strategies in diseases associated with infections and mucosal surface dehydration.

## Figures and Tables

**Figure 1 cells-11-01587-f001:**
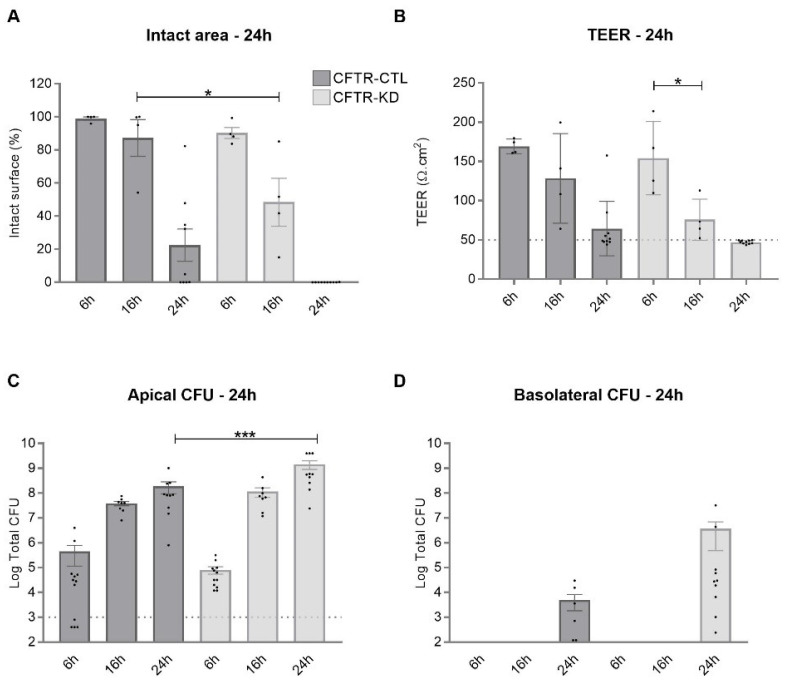
Vulnerability of the CFTR-KD epithelium to WT PAO1. The CFTR-CTL (dark gray) and CFTR-KD (light gray) epithelia were apically infected with 10^3^ CFU of the wild type PAO1 strain for 6 h*,* 16 h, and 24 h. The intact epithelial surface areas (**A**), TEER values (**B**), apical (**C**), and basolateral (**D**) CFUs were measured in uninfected and infected conditions. (**A**) The intact epithelial surface was expressed as % of the area measured in uninfected conditions (6 h: *N* = 4, *n* ≥ 1; 16 h: *N* = 4, *n* = 1; 24 h: *N* ≥ 9, *n* = 1). (**B**) TEER values, which were measured in duplicate for each insert, were expressed as Ω.cm^2^. The dotted line indicates the average TEER values of empty inserts (6 h: *N* = 4, *n* ≥ 2; 16 h: *N* = 4, *n* = 2; 24 h: *N* = 10, *n* = 2). (**C**) The dotted line represents the initial amount of inoculated bacteria (6 h: *N* = 4, *n* ≥ 2; 16 h: *N* = 4, *n* = 2; 24 h: *N* = 10, *n* ≥ 2). *, *** indicate the degree of significance and two-way ANOVA tests, respectively. (**D**) (6 h: *N* = 4, *n* = 2; 16 h: *N* = 4, *n* = 2; 24 h: *N* = 10, *n* ≥ 2). 10^2^ CFU/well is the detection limit. Points with a value of zero do not appear in logarithmic scale.

**Figure 2 cells-11-01587-f002:**
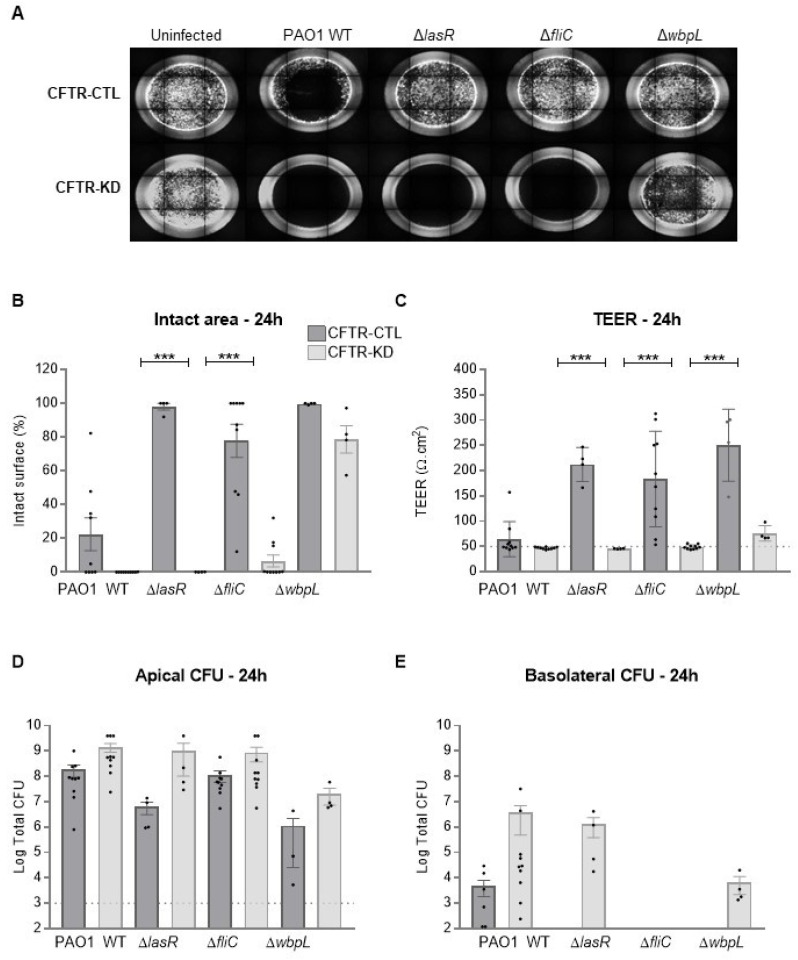
Vulnerability of the CFTR-KD epithelium to PAO1 strains with attenuated virulence at 24 h post-infection. CFTR-CTL (dark gray) and CFTR KD (light gray) epithelia were apically infected with 10^3^ CFU of the wild type PAO1 strain or PAO1 mutants (Δ*lasR*, Δ*fliC*, Δ*wbpL*) for 24 h. The intact epithelial surface (**A**,**B**), TEER values (**C**), apical (**D**), and basolateral (**E**) CFUs were determined in uninfected and infected conditions. (**A**) Representative images of lesions induced by each bacterial strain in the CFTR-CTL and CFTR-KD epithelia. (**B**) The intact epithelial surface was expressed as % of the initial area measured in uninfected conditions (PAO1 WT: *N* ≥ 9, *n* = 1; Δ*lasR*: *N* = 4, *n* = 1; Δ*fliC*: *N* = 10, *n* = 1; Δ*wbpL*: *N* = 4, *n* = 1). (**C**) TEER values, which were measured in duplicate for each insert, were expressed as Ω.cm^2^. The dotted line indicates the average TEER values of empty Transwell filters (PAO1 WT: *N* ≥ 9, *n* = 2; Δ*lasR*: *N* = 4, *n* = 2; Δ*fliC*: *N* = 10, *n* = 2; Δ*wbpL*: *N* = 4, *n* = 2). (**D**) The dotted line represents the initial amount of inoculated bacteria. *** indicates the degree of significance, two-way ANOVA tests (PAO1 WT: *N* ≥ 9, *n* ≥ 2; Δ*lasR*: *N* = 4, *n* ≥ 2; Δ*fliC*: *N* = 10, *n* ≥ 2; Δ*wbpL*: *N* = 4, *n* ≥ 2). (**E**) (PAO1 WT: *N* ≥ 9, *n* ≥ 2; Δ*lasR*: *N* = 4, *n* ≥ 2; Δ*fliC*: *N* = 10, *n* ≥ 2; Δ*wbpL*: *N* = 4, *n* ≥ 2). Points with a value of zero do not appear in logarithmic scale.

**Figure 3 cells-11-01587-f003:**
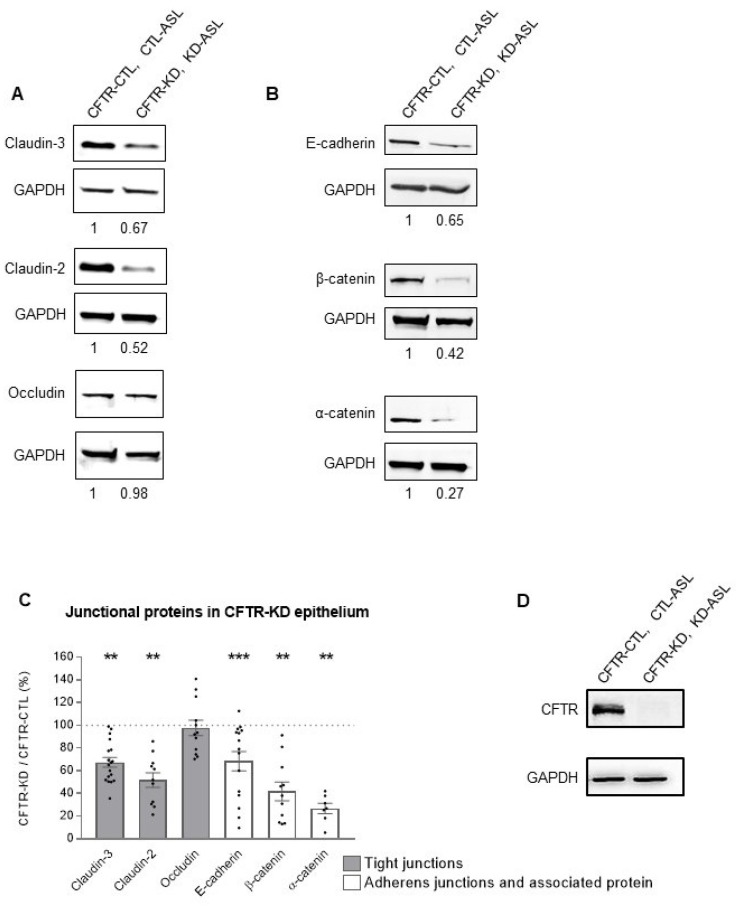
Altered expression of junctional proteins in the CFTR-KD epithelium. Junctional protein expressions were evaluated by Western blot. Representative Western blots for tight-junction (**A**) and adherens-junction (**B**) proteins. The relative expression level is indicated below each lane. (**C**) Quantification of protein expression in the CFTR-KD epithelia relative to their expression in the CFTR-CTL cells (dotted line). **, *** indicate the degree of significance, paired Student tests, *N* ≥ 7, *n* ≥ 1. (**D**) Western blot confirming the CFTR protein knockdown in the CFTR-KD cell compared to the CFTR-CTL cells.

**Figure 4 cells-11-01587-f004:**
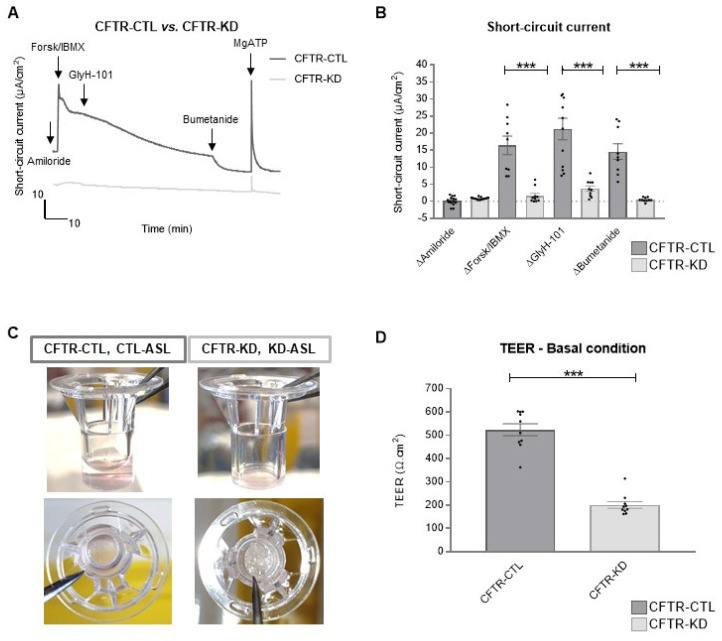
Defective CFTR-dependent ion transport, ASL volume, and transepithelial resistance of the CFTR-KD epithelium. (**A**) The CFTR-CTL and CFTR-KD Transwell inserts were bathed between an apical and a basal solution creating a Cl^-^ gradient, continuously gassed with 95% O_2_ and 5% CO_2_ at 37 °C. The potential difference was clamped to monitor the short-circuit current value (Isc, µA/cm^2^) during the addition of amiloride (apical 100 µM); forskolin/IBMX (apical and basal 10/100 µM); GlyH-101 (apical 20 µM); bumetanide (basal 100 µM); MgATP (basal 100 µM); and DMSO as the control of the GlyH-101 effect. Representative traces of the CFTR-CTL (dark gray) and CFTR-KD (light gray) Isc currents are shown. (**B**) Mean ± SEM ΔIsc currents evoked by the different drugs in the CFTR-CTL and CFTR-KD (*N* ≥ 5, *n* = 2) inserts. *** indicates the degree of significance, two-way ANOVA tests. (**C**) Representative images of the CFTR-CTL and CFTR-KD epithelia polarized for at least 15 days on Transwell filters at ALI showing the low ASL volume and dehydrated surface of the CFTR-KD cultures. (**D**) TEER values (Ω.cm^2^) of the CFTR-CTL (dark gray) and CFTR-KD (light gray) cells polarized for at least 15 days on Transwell filters at ALI (*N* = 10, *n* ≥ 2). *** indicates the degree of significance, paired Student tests.

**Figure 5 cells-11-01587-f005:**
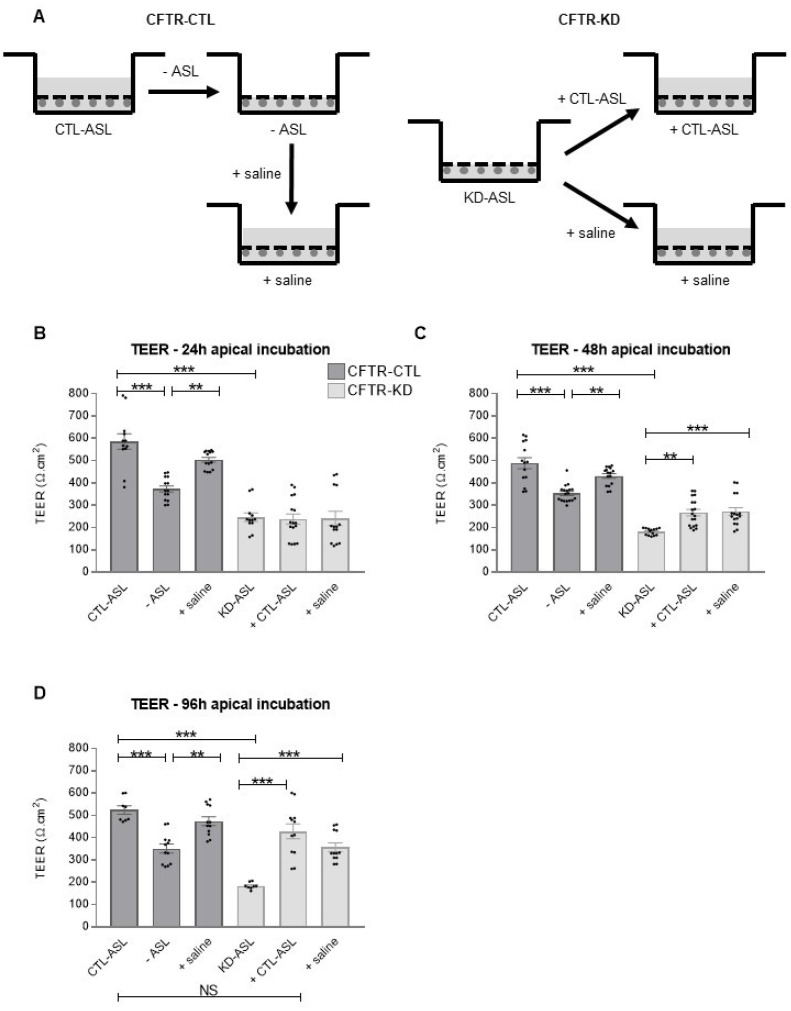
Restoration of the TEER values with rehydration of the CFTR-KD epithelium surface. (**A**) Scheme of the ASL manipulations performed in the CFTR-CTL and CFTR-KD Transwell inserts. The CTL-ASL of the CFTR-CTL cultures (dark gray) was kept (CTL-ASL condition), removed (-CTL-ASL condition), or removed and replaced by the same amount of physiological saline (+ saline condition) for 24 h (*N* = 3, *n* ≥ 4) (**B**), 48 h (*N* = 4, *n* ≥ 2) (**C**), and 96 h (*N* = 3, *n* ≥ 2) (**D**). For the CFTR-KD cultures (light gray), CTL-ASL (collected from CFTR-CTL cultures, + CTL ASL condition) or the same amount of physiological saline (+saline condition) was added or not to the KD-ASL for 24 h (*N* = 3, *n* ≥ 4) (**B**), 48 h (*N* = 4, *n* ≥ 2) (**C**), or 96 h (*N* = 3, *n* ≥ 2) (**D**). After the 24 h period, the TEER values (Ω.cm^2^) were measured in duplicate. **, *** indicate the degree of significance, two-way ANOVA tests.

**Figure 6 cells-11-01587-f006:**
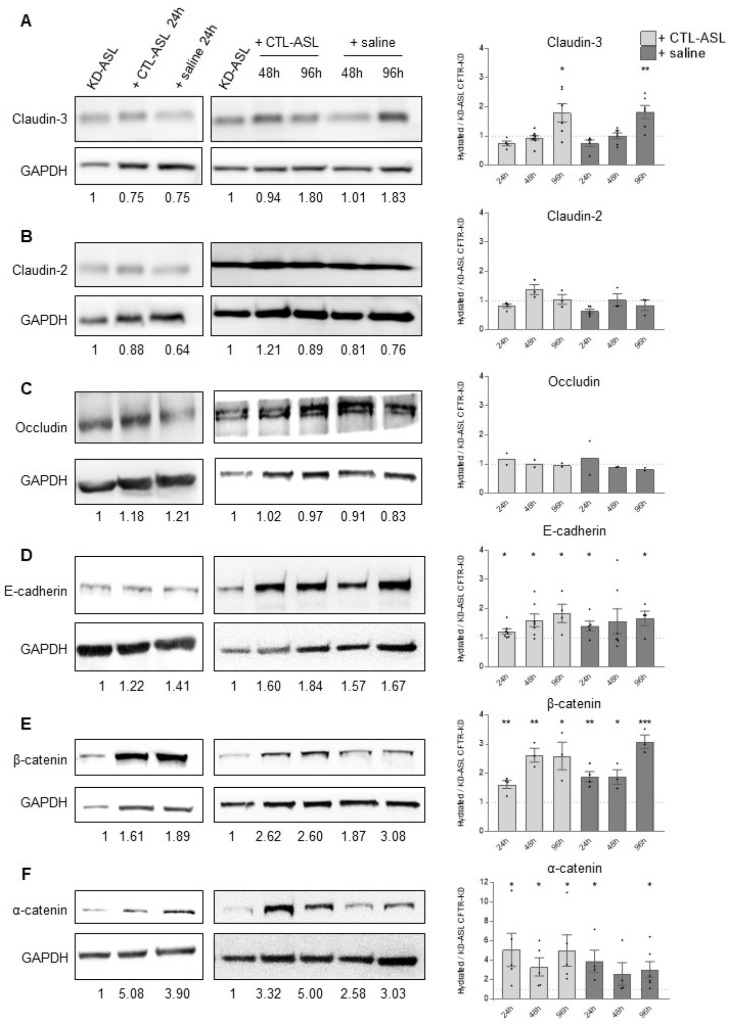
Expression of junctional proteins in the apically rehydrated CFTR-KD epithelium. The amount of junction protein expression was estimated by Western blot and quantified in the CFTR-KD epithelia before and 24 h (left panels), 48, and 96 h (right panels) after apical rehydration with CTL-ASL or physiological saline. Representative Western blot for claudin-3 (*N* ≥ 4, *n* = 1) (**A**), claudin-2 (*N* ≥ 3, *n* = 1) (**B**), occludin (*N* = 2, *n* = 1) (**C**), E-cadherin (*N* ≥ 4, *n* = 1) (**D**), β-catenin (*N* ≥ 3, *n* = 1) (**E**), and α-catenin (*N* ≥ 4, *n* = 1) (**F**). The relative expression level is indicated below each lane. *, **, *** indicate the degree of significance, paired Student tests. Note the different scale for the α-catenin protein expression.

**Figure 7 cells-11-01587-f007:**
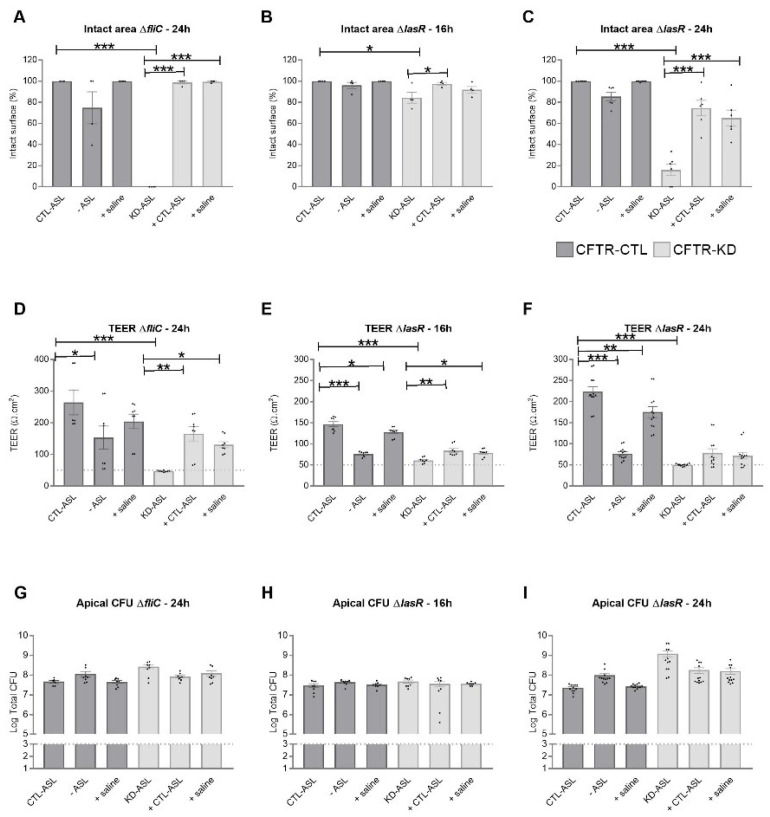
Protection of the CFTR-KD epithelium to PAO1 strains with attenuated virulence by rehydration. CFTR-CTL (dark gray) and CFTR-KD (light gray) epithelia were apically infected with 10^3^ CFU of Δ*fliC* (**A**,**D**,**G**) or Δ*lasR* for 16 h (**B**,**E**,**H**) or 24 h (**C**,**F**,**I**). The intact epithelial surface (**A**–**C**), TEER values (**D**–**F**), and apical (**G**–**I**) CFUs were determined in response to ASL manipulation. (**A**–**C**) The intact epithelial surface was expressed as % of the initial area measured in uninfected conditions (Δ*fliC*: *N* = 3, *n* = 1; Δ*lasR* 16 h: *N* = 4, *n* = 1; Δ*lasR* 24 h: *N* = 6, *n* = 1). (**D**–**F**) TEER values (Ω.cm^2^) were measured in duplicate for each insert. The dotted line indicates the average TEER values of empty Transwell filters (Δ*fliC*: *N* = 3, *n* = 2; Δ*lasR* 16 h: *N* = 4, *n* = 2; Δ*lasR* 24 h: *N* = 6, *n* = 2). (**G**–**I**) The dotted line represents the initial amount of inoculated bacteria (Δ*fliC*: *N* = 3, *n* = 2; Δ*lasR* 16 h: *N* = 4, *n* = 2; Δ*lasR* 24 h: *N* = 6, *n* = 2). *, **, *** indicate the degree of significance, two-way ANOVA tests.

**Figure 8 cells-11-01587-f008:**
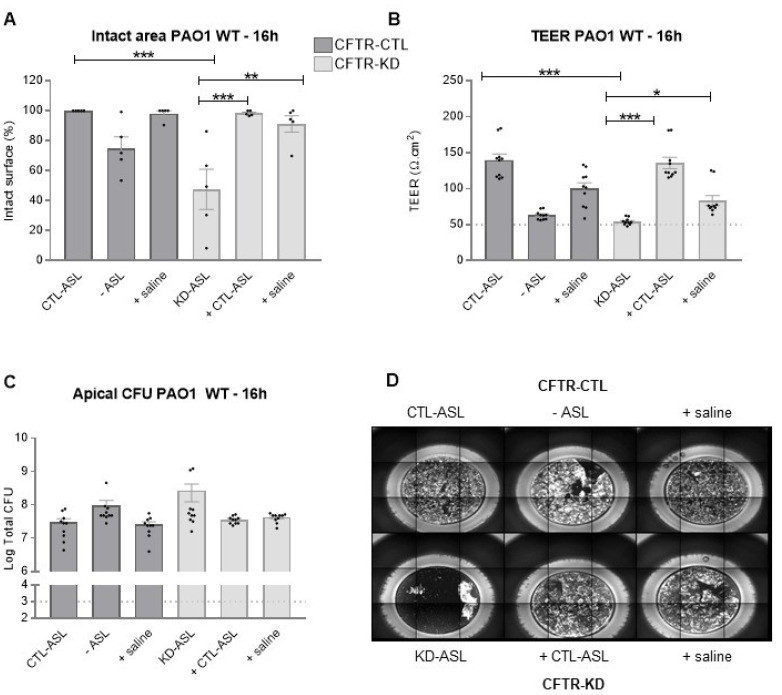
Protection of the CFTR-KD epithelium to the WT PAO1 strain by rehydration. CFTR-CTL (dark gray) and CFTR-KD (light gray) epithelia were apically infected with 10^3^ CFU of WT PAO1 for 16 h. The intact epithelial surface (**A**), TEER values (**B**), and apical (**C**) CFUs were determined in response to ASL manipulation. (**A**) The intact epithelial surface was expressed as % of the initial area measured in uninfected conditions (*N* = 5, *n* = 1). (**B**) TEER values (Ω.cm^2^) were measured in duplicate for each insert. The dotted line indicates the average TEER values of empty Transwell filters (*N* = 5, *n* = 2). (**C**) The dotted line represents the initial amount of inoculated bacteria (*N* = 5, *n* = 2). *, **, *** indicate the degree of significance, two-way ANOVA tests. (**D**) Representative images of lesions induced by WT PAO1 in the CFTR-CTL and CFTR-KD epithelia.

## Data Availability

The dataset for this article can be found at the following DOI: 10.26037/yareta:ckvglc7mgbdrjml5k6nllgtlhe. It will be preserved for 10 years.

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
