# Peer review of "Surface Hydration Protects Cystic Fibrosis Airways from Infection by Restoring Junctional Networks"

_cells, 2022, doi:10.3390/cells11091587_

Round 1
Reviewer 1 Report
In the manuscript entitled „Surface hydration protects cystic fibrosis airways from infection by restoring junctional networks”, Juliette Simonis and colleagues use CFTR-deficient airway epithelial cells to analyze the impact of airway surface liquid (ASL) on the integrity of the epithelial layer, its junctional proteins and its susceptibility to P. aeruginosa infections in the context of cystic fibrosis. Interestingly, the authors are able to demonstrate that experimental rehydration of CFTR-deficient epithelial cells was able to induce the expression of key junctional proteins (e.g., α-catenin, β-catenin, E-cadherin and claudin-3), to avoid the loss of epithelial integrity and to decrease the level of P. aeruginosa-induced cytotoxicity. Moreover, the described findings indicate that the observed interplay between ASL and epithelial integrity seems to be mostly independent from the molecular composition of the ASL, as comparable effects could be described after the apical addition of physiological saline or control epithelial cell-derived ASL. Overall the presented results provide new insights into a so far less considered partial aspect of CF pathogenesis, indicate that the presence of sufficient apical liquid volume is required for the maintenance of epithelial airway homeostasis and imply that therapeutic rehydration of CF airway epithelium may not only mean a relevant benefit for mucociliary transport, but might also directly reinforce epithelial barrier integrity.
Although the comprehensibly described data acquired in well-designed and extensive cell culture experiments are interesting, the current version of the manuscript shows the following limitations, which relevantly restrict its overall impact:
- The in vivo relevance of the here in vitro in cultured epithelial cells identified impact of the apical liquid on the epithelial barrier integrity and the regulation of junctional proteins should either be confirmed experimentally or, at least, discussed more in detail.
- The authors only speculate about an epithelial integrity “gate-keeper”, which might be modulated by the presence of an apical liquid volume, but they don’t identify this gate-keeper experimentally or provide/discuss any precise mechanism by which the ASL-mediated regulation of the expression of junctional proteins can be explained on a molecular level.
Moreover, the following aspects should be addressed by the authors:
- Based on the data depicted in Figure 1, the authors conclude that CFTR-deficient epithelial cells are more vulnerable to wild type PAO1 infection and show an earlier loss of barrier integrity. However, in Fig. 1A and Fig. 1D, the observed differences in wild type PAO1-induced epithelial damage and the basolateral bacterial content between CFTR-deficient and control epithelial cells do not appear to be statistically significant (at least statistical significance has not been indicated in the respective graphs), which somehow weakens the functional relevance of the observed phenomenon. Regarding the graph depicted in Fig. 1B, the value of the column for CFTR-CTL at 16h does not seem to represent the mean of the depicted three single data points; only three data points are depicted, although N=4 experiments have been performed due to the figure legend. This seems to also be the case for CFTR-KD at 6h. In general, the authors should check carefully for all graphs presented in the manuscript whether all single data points used for the calculation of the respective mean values are depicted in the graphs.
- In Fig. 6, the CTL-ASL- or saline-induced increase in α-catenin, β-catenin, E-cadherin and claudin-3 expression in CFTR-KD cells does not reach statistical significance (at least statistical significance has not been indicated in the respective graphs), which makes it difficult to draw a definitive conclusion from these data.
Reviewer 3 Report
Simonin et al. performed a impressive work on the role of ASL in the protection of respiratory epithelium against Pseudomonas aeruginosa in CF Calu-3 cultures. I take pleasure to read the manuscript and I have some comments which I think could improve it.
Methods:
- I am not sure to understand how you can collect 100µl of apical medium for CFU counts as you add only 10µL of bacterial solution and as calu-3 cells produced +/- 30µl of ASL after 2 wks of ALI culture? Did you add 100µL of saline to wash the apical side of the cells?
- Can you clarify how you quantify the percentage of intact area by microscopy with ImageJ? How many fields are analyzed? How did measure the holes?
Results:
- I think that the manuscript would be easier to follow if the authors begin first with the characterization of their cellular model (CFTR-KD versus control: TEER, junctional protein expressions) (Figure 3,4), secondly with ASL/saline protective role (Figure 5,6) and finally the consequences of Pseudomonas aeruginosa infection (Figure 1,2,7).
- Can you present your data for TEER in the same way on each graph, in Ω.cm²?
- Can you show the picture of the inserts for each infection experiments before the quantification (like in Figure 2)?
- Can you discuss the discrepancies between the different junctional protein expressions (e.g. the absence of decrease of occludin) in Figure 3 and in Figure 6.
- I would confirm the junctional protein expression by RT-qPCR.
- I would perform ZO-1 WB to confirm immunofluorescence data.
- Can you confirm the protective role of ASL/saline by showing that there is no/less bacterial growth at 24hrs in the CFTR-KD epithelium basolateral medium than in basal conditions (in Figure 1)?
- Can you more precisely explain which panels of genes are explored with the nanospring? Are the upregulated/downregulated genes the same in controls and CF calu-3 in the different conditions?
Discussion:
- You discuss the role of hydration of the respiratory epithelium in people with CF. Is there some data showing the action of hypertonic saline 3-6-7% on ASL in epithelial cultures? Can you maybe try to treat the CF Calu-3 cells to see if it is more efficient to restore TEER or intact area than saline 0,9%?
Minor comments:
Line 195: inoculation of 103 CFU of wild -> inoculation of 10³ CFU of wild
Round 2
Reviewer 1 Report
The authors have sufficiently addressed my former concerns regarding Fig. 1 and Fig. 6.
Formally, the authors also answered my questions about the in vivo relevance of their results obtained in the cell culture system and a possible mechanism underlying the ASL-mediated regulation of the expression of junctional proteins; but in both these points, the quality of their reply remained slightly behind my expectations. In principle, the authors indicate that addressing these two questions experimentally, while interesting, is beyond the scope of the current manuscript. Although this decision somehow limits the overall impact of the manuscript, it is acceptable in principle. However, under this condition I would have hoped that the authors would have succeeded even better in supporting the expected in vivo relevance of their results on the basis of literature and in discussing even more precise signaling cascades potentially linking ASL-mediated regulation of the expression of junctional proteins. In my opinion, just mentioning the fact that in vivo confirmation is awaited, although correct and important, cannot further strengthen the clinical implications of the manuscript.
Reviewer 3 Report
No more comment.